# Mother’s Own Milk Feeding in Preterm Newborns Admitted to the Neonatal Intensive Care Unit or Special-Care Nursery: Obstacles, Interventions, Risk Calculation

**DOI:** 10.3390/ijerph18084140

**Published:** 2021-04-14

**Authors:** Nadja Heller, Mario Rüdiger, Vanessa Hoffmeister, Lars Mense

**Affiliations:** Department of Pediatrics, Division of Neonatology & Pediatric Intensive Care, Saxonian Center for Feto/Neonatal Health, University Hospital Carl Gustav Carus Dresden, TU Dresden, 01307 Dresden, Germany; nadja.heller84@gmail.com (N.H.); mario.ruediger@uniklinikum-dresden.de (M.R.); Vanessa.hoffmeister@tu-dresden.de (V.H.)

**Keywords:** neonatal nutrition, breast feeding, nutrition/growth, mother’s own milk

## Abstract

Early nutrition of newborns significantly influences their long-term health. Mother’s own milk (MOM) feeding lowers the incidence of complications in preterm infants and improves long-term health. Unfortunately, prematurity raises barriers for the initiation of MOM feeding and its continuation. Mother and child are separated in most institutions, sucking and swallowing is immature, and respiratory support hinders breastfeeding. As part of a quality-improvement project, we review the published evidence on risk factors of sustained MOM feeding in preterm neonates. Modifiable factors such as timing of skin-to-skin contact, strategies of milk expression, and infant feeding or mode of delivery have been described. Other factors such as gestational age or neonatal complications are unmodifiable, but their recognition allows targeted interventions to improve MOM feeding. All preterm newborns below 34 weeks gestational age discharged over a two-year period from our large German level III neonatal center were reviewed to compare institutional data with the published evidence regarding MOM feeding at discharge from hospital. Based on local data, a risk score for non-MOM feeding can be calculated that helps to identify mother–baby dyads at risk of non-MOM feeding.

## 1. Introduction

Breast-milk feeding has significant advantages for infants. Premature infants receiving their mother’s own milk (MOM) have a lower incidence of necrotizing enterocolitis [1], sepsis [2], and retinopathy of prematurity than formula-fed preterm infants [3]. In the long term, MOM reduces the risk of bronchopulmonary dysplasia [4] and metabolic syndrome [5]. Despite suboptimal weight gain during their hospital stay, breastfed preterm infants had advantages in neurodevelopmental assessment at two and five years of age [6]. Breastfeeding is also advantageous for mothers: It reduces the risk of breast and ovarian cancer [7] and postpartum bleeding, and promotes uterine involution [8]. The costs for institutions to supply breast milk during hospitalization are relatively low in comparison to other healthcare-related costs [9].

Thus, the World Health Organization (WHO) and the American Academy of Pediatrics (AAP) recommend exclusive breastfeeding for the first six months [10].

Prematurity with hospitalization after birth, however, raises barriers for an adequate breast-milk utilization: Mother and child are separated to allow necessary medical treatment of the infant. Due to immaturity, sucking is poor, and expression of breast milk is necessary for several weeks before the infant can be breastfed. Several other risk factors decreasing the likelihood of MOM feeding at discharge from hospital have been described previously and will be reviewed subsequently. While some risk factors are modifiable by unit policies, many are not, since they are of a demographical nature. Nevertheless, both modifiable and unmodifiable risk factors are important to consider when targeted interventions are implemented for those mother–baby dyads, which are at the highest risk for NMOM (non-mother’s own milk; i.e., formula or donor milk) feeding at discharge.

During the last decades, the rate of breast-milk feeding has substantially increased in preterm infants. For example in France, breast-milk-feeding rates increased from 19% in 1997 [6] to 47% in 2011 [11], but the variability between units and countries is enormous, and suggests room for systematic improvements [12].

In-depth analyses of discharge feedings in German neonatal intensive care units (NICUs) have rarely been published. As part of a quality-improvement project, we summarized published evidence and compared our local data regarding MOM feeding in preterm infants at discharge with the literature, and will present the results accordingly. MOM feeding is defined as exclusive breast-milk feeding during the last 48 h prior to discharge from hospital, either by breastfeeding or as extracted breast milk (EBM) by bottle. The odds ratios of influencing factors of MOM feeds were calculated. Additional information on statistics can be found in Appendix A.

In our institution, 368 preterm newborns below 34 weeks gestational age were discharged between 1 April, 2013 and 31 March, 2015 from the NICU or special-care nursery (SCN) of our level III neonatal center. Of those, 94% were inborn and 6% were admitted from a community hospital within 48 h after birth. Most newborns were admitted to the NICU initially (97.6%). Patients were born at 30^6/7^ (28^3/7^; 32^3/7^) weeks and discharged at 36^0/7^ (35^0/7^; 37^0/7^) weeks corrected gestational age.

It is hospital policy to encourage mothers to use electrical milk pumps, which are available in the hospital and are prescribed for use at home after the mother is discharged. Extracted breast milk is stored in a central hospital facility and thawed for each use. Hospital policies on preterm enteral feeding and breast-milk expression remained unchanged throughout the study period.

Extracted breast milk requires fortification to meet the nutritional needs of preterm infants. In our institution, fortification is initiated once the volume of enteral feeds reaches 120 mL/kg/day and is continued until discharge from hospital or due date.

A total of 60.1% of the patients were discharged on exclusive MOM feeds. This rate was higher than in the EPIPAGE-2 cohort study, which reported 25% of exclusive and 47% of some MOM feeds at discharge in infants below 32 weeks [11].

## 2. Modifiable Risk Factors for NMOM Feeding at Discharge

### 2.1. Skin-to-Skin Contact in the Delivery Room and the NICU

Skin-to-skin contact (SSC), or kangaroo care, describes placing the undressed newborn on the bare chest of one of the parents. SSC is an important component of family-centered care and is advised by the WHO, the Baby Friendly Hospital Initiative, and other organizations. Positive effects of SSC have been described on the infant’s cardiovascular and respiratory stability, self-regulation, weight gain, etc. Confidence and bonding with the baby are benefits for the parents. Delayed or absent SSC has been described as a risk factor of NMOM feeding [11,13,14,15]. Notably in one study, longer daily hours of SSC were associated with earlier exclusive breastfeeding in preterm infants born at 28 to 34 weeks who spent a median daily duration of 7.5 h in SSC [13].

In our institution, SSC in the delivery room was not associated with MOM feeds at discharge, but SSC during the first week of life, outside of the delivery room, was (Table 1).

First, SSC in the NICU was initiated later in the NMOM group at 7 (4; 12) days of life than in the MOM group, where it was initiated at 5 (3; 10) days of life (*p* = 0.02).

### 2.2. Strategies of Infant Feeding and Milk Expression

Policies to support the initiation and maintenance of breast-milk feeding can increase the rates of MOM feeding at discharge [11]. Those policies might include information on breast-milk feeding given to moms admitted to the hospital for threatened preterm labor (nonsignificant effect) and the proposal of breast-milk expression within 6 h after birth (significant effect). Furthermore, policies for MOM-feeding maintenance include unit policies for human-milk or breast-milk feeding (significant effect) and a special room for mothers to pump their milk (nonsignificant effect) [11]. A delay of the first breast-milk expression for more than 48 h postpartum is an important risk factor for NMOM feeding [16].

The employment of professionals trained in human lactation (either according to the International Board Certified Lactation Consultant or French master training in human lactation) did not increase MOM feeding rates at discharge in the French EPIPAGE-2 study [11], although it increased breast-milk initiation [17]. In other studies, training of healthcare workers in hospitals did increase MOM feeding, even after discharge [18].

The use of nipple shields for breastfeeding is associated with less MOM feeding at discharge in preterm infants, and should therefore be critically appraised [16,19]. The motives for nipple-shield use in the extremely preterm infant were more often related to the infant than to the mother. The use for infants who fell asleep at the breast was especially associated with NMOM feeding at discharge. Only nipple shields for too-engorged breasts increased MOM feeding [19]. Minimizing the use of pacifiers lowered the odds of NMOM feeding as well [16].

In our institution, 74.7% of infants were breastfed at least once during their hospital stay. In infants <28 weeks gestational age, breastfeeding was initiated significantly later, at a corrected gestational age of 34^2/7^ (32^3/7^; 36^2/7^) weeks, than in infants at or above 28 weeks, where it was initiated at 33^4/7^ (32^5/7^; 34^1/7^) weeks (*p* = 0.047).

The rate of MOM feeds at discharge was significantly higher in infants who were breastfed at least once during their hospital stay (Table 2). In infants <28 weeks, 30/41 (73.2%) were on MOM at discharge if they were breastfed once, while only 6/36 (16.7%) of those never being breastfed received MOM, OR 13.64 (95% CI 4.47–41.63). Nipple shields are not routinely utilized in our unit, and their use is not recorded.

### 2.3. Mode of Delivery

The initiation of breastfeeding is hampered by a caesarean section in term newborns [20,21,22]. Planned C-sections are especially associated with breastfeeding cessation before 12 weeks postpartum [23]. Unfortunately, data from the preterm population is scarce.

In our institution, most preterm newborns are born via C-section. Vaginal delivery showed a trend toward more MOM feeding at discharge, but did not reach statistical significance (Table 3).

## 3. Unmodifiable Risk Factors for NMOM Feeding at Discharge

### 3.1. Demographical Risk Factors of the Infant

Prematurity itself and low birth weight are risk factors for NMOM feeding: Newborns <28 weeks are at an 2.9-times increased risk of exclusive breastfeeding failure at discharge [16]. In an Italian study, no human milk was fed to 45% of infants <1500 g birth weight, while only 23% of infants >2500 g received no human milk at discharge [20]. Data on multiple births is controversial, since some studies demonstrate an association with exclusive breastfeeding [21], while others show an association with NMOM feeds [16,22] or discontinuation of breastfeeding before six months of life [23].

In our institution, MOM feeds were significantly more common in infants of 28 weeks and above than in those below 28 weeks (Table 4): The rate of exclusive formula feeding decreased with higher gestational age at birth (Figure 1). Multiples were not at higher risk of NMOM feeds than singletons in our cohort. Birth weight was not associated with the likelihood of NMOM feeds.

### 3.2. Demographical Risk Factors of the Mother

Young maternal age is a risk factor for NMOM feeding at discharge. In preterm infants <34 weeks, each year of maternal age was associated with a 1.24-times increase of direct breastfeeding at hospital discharge [24]. Mothers <25 years of age ceased breastfeeding more often before discharge and before six months of age than mothers older than 25 years [23,25].

In studies in the U.S., marital status is often assessed. In term newborns, married women initiated and maintained breastfeeding longer than single mothers [26]. Conflicting results exist for preterm newborns <30 to 32 weeks. While in one study, unmarried women initiated breast-milk feeds more often, and more unmarried mothers provided breast milk until discharge [21], in another study infants from married mothers were often on MOM at discharge [27]. European studies rarely mention marital status, thus no data on European newborns is available.

Maternal socioeconomic status influences breast-milk feeding in term and preterm newborns: More mothers of lower educational level or on social welfare cease breastfeeding before discharge [22] or before six months of age [28] in European countries. Large regional differences in breastfeeding rates of preterm infants at discharge and at six months of age are common among European countries [23,25].

Breastfeeding experience in a previous offspring has protective effects on breastfeeding in the NICU: Mothers who did not breastfeed previous infants have 5.6-times higher odds to discontinue exclusive breastfeeding before discharge than those with at least four months of breastfeeding experience [16]. The support of the mother’s partner in supplying breast milk is promoting MOM feeding [27].

In our institution, maternal age, an academic degree, or parity was not associated with different rates of MOM feeding at discharge (Table 5).

### 3.3. Health and Disease-Related Risk Factors of the Infant

Infants with bronchopulmonary dysplasia (BPD) have significantly lower odds of being breastfed at discharge [25] and at six months of age [23]. The influence of other neonatal comorbidities has rarely been studied, although an effect of abdominal surgeries and other complications on MOM feeding seems possible. Severe morbidity, defined as culture-proven sepsis, necrotizing enterocolitis, or focal intestinal perforation, was not associated with NMOM feeds at discharge in newborns <1500 g, but length of stay on the NICU clearly was [29]. Severe morbidity, defined as retinopathy of prematurity, intraventricular hemorrhage, periventricular leukomalacia, and necrotizing enterocolitis, did not affect breastfeeding continuation in newborns below 32 weeks at six months of age [23].

In our institution, intraventricular hemorrhage (grade 2 or higher) and posthemorrhagic ventricular dilatation did not affect MOM feeding at discharge. Infants with any kind of abdominal surgery were less likely to be discharged on MOM (Table 6).

### 3.4. Health and Disease-Related Risk Factors of the Mother

Mothers with nicotine abuse during pregnancy rarely give MOM feeds to their preterm infants at discharge (odds ratio of 0.43 (0.25–0.76) in a Danish study) [16,22,24]. Among women who stopped smoking for the pregnancy, those who relapsed to daily smoking postpartum were most likely to discontinue breastfeeding [30]. Recreational drug use has similar effects on direct breastfeeding of preterm infants [24].

Gestational diabetes decreases the odds for exclusive breastfeeding at hospital discharge in a study of mostly term newborns [31]. The underlying mechanisms were unclear, but delayed lactogenesis and the interference of early supplemental feeds with breastfeeding were discussed. Obesity can delay lactogenesis as well, and is associated with a shorter breastfeeding period. Lack of confidence and body comfort also shortened the duration of breastfeeding [32].

In our institution, the use of nicotine or recreational drugs was significantly associated with less likely MOM feeding at discharge. Gestational diabetes also was associated with fewer MOM feeds (Table 7).

### 3.5. Maternal Experiences with Milk Expression

The large majority of mothers of preterm children describe human-milk feeding as beneficial and important for their children, and they perceive supportive attitudes of hospital staff. Nevertheless, experiencing problems and barriers when pumping breast milk postpartum is a regular phenomenon for mothers of preterm children [33]. Infants of mothers who described difficulties in pumping breast milk or providing an adequate amount of breast milk for their child had a higher likelihood of receiving formula exclusively at discharge (OR 4.6, CI 1.5–13.9; and OR 3.6, CI 1.1–11.5, respectively) [33]. Achieving a volume of 500 mL per day of pumped breast milk by day 14 postpartum is a strong indicator of MOM feeding at discharge in newborns <1500 g (OR 9.7, CI 3.9–24.4) [34].

## 4. Identifying Mother–Baby Dyads at Risk of NMOM Feeding at Discharge

We aimed to identify parameters that allow an early prediction of the risk to be on NMOM at discharge. The risk score can be calculated at the end of the first week of life and includes three parameters: (I) gestational age below 28 weeks at birth, (II) no skin-to-skin care until the end of the first week of life, and (III) maternal use of alcohol or recreational drugs. One point is given for each parameter.

The percentage of MOM feeds at discharge in our cohort was 69.1% (0 points), 61.3% (1 point), 41.9% (2 points), and 12.5% (3 points) (Table 8). We suggest implementation of special measures to support families in continued MOM feeding if the risk score is 2 or higher, which applied to 82/368 newborns of our population (22.2%).

The risk score is easy to calculate, and all information required is available at the end of the first week of life. It is based on our local data set, and should be evaluated in another cohort of preterm newborns in the future. A mother–baby dyad with a low risk score might still benefit from breastfeeding support, but limited resources can be allocated to mothers at higher risk. Even in the lowest risk group, 30.9% of the preterm newborns were on NMOM at discharge.

## 5. Checklist for Quality-Improvement Projects to Increase MOM Feeding at Discharge

MOM feeding rates can be increased by regular quality assessment and implementation of policies. The following checklist might support divisional assessments:Is MOM feeding at discharge seen as an important outcome parameter?Is skin-to-skin care in the delivery room facilitated?Is skin-to-skin care in the NICU possible during the first days of life, especially in the first week of life?Is breast-milk expression started within 6 h after delivery?Do parents receive information on breastfeeding/breast-milk expression antenatally?Is the amount of pumped breast milk discussed at daily rounds?Are targeted interventions available for infants at risk, such as after abdominal surgery or below 28 weeks gestational age?Are targeted interventions available for mothers at risk, such as with nicotine use or gestational diabetes?Are targeted interventions available for mothers not achieving 500 mL per day of pumped breast milk at day 14 postpartum?

## 6. Conclusions

Mom’s own milk feeding is beneficial for preterm infants, but several modifiable and unmodifiable factors influence feeding at discharge from the NICU. Appraising unmodifiable risk factors allows care providers to initiate interventions to promote sustained MOM feeding on an individual basis. Modifiable risk factors can be targeted by institutional policies and should be reviewed accordingly.

## Figures and Tables

**Figure 1 ijerph-18-04140-f001:**
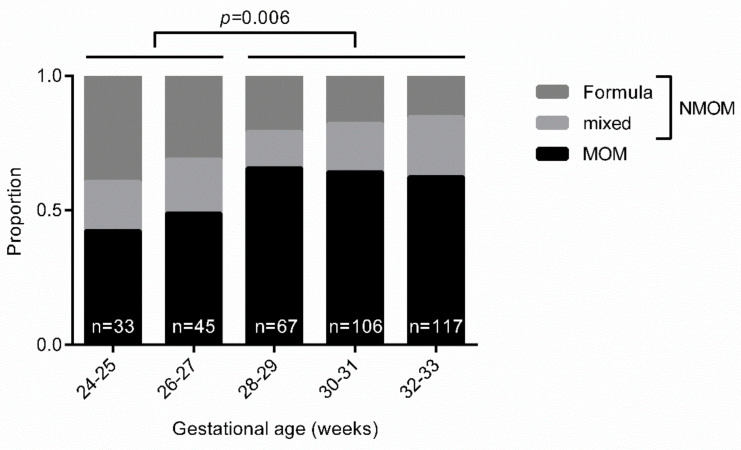
Feeding at discharge. Significantly more infants of 28 weeks gestational age and above are fed MOM at discharge than below 28 weeks.

**Table 1 ijerph-18-04140-t001:** SSC (Skin-to-skin contact) and MOM (Mother’s own milk) feeding at discharge.

Modifying Factor		MOM Feeding at Discharge	
	Total	N (%)	OR (95% CI)
SSC in the delivery room			
No	267	160 (60.0)	1.00 (Reference)
Yes	54	38 (70.4)	1.59 (0.84–2.99)
SSC in the first week of life			
No	169	90 (53.3)	1.00 (Reference)
Yes	199	133 (65.8)	1.69 (1.11–2.58)

**Table 2 ijerph-18-04140-t002:** Breastfeeding and MOM feeding at discharge.

Modifying Factor		MOM Feeding at Discharge	
	Total	N (%)	OR (95% CI)
Breastfeeding at least once during hospital stay			
No	92	26 (28.3)	1.00 (Reference)
Yes	275	195 (70.9)	6.19 (3.67–10.44)

**Table 3 ijerph-18-04140-t003:** Mode of delivery and MOM feeding at discharge.

Modifying Factor		MOM Feeding at Discharge	
	Total	N (%)	OR (95% CI)
Mode of delivery			
Vaginal delivery	56	40 (71.4)	1.00 (Reference)
C-section	312	181 (58.0)	0.55 (0.30–1.03)

**Table 4 ijerph-18-04140-t004:** Infant characteristics and MOM feeding at discharge.

Modifying Factor		MOM Feeding at Discharge	
	Total	N (%)	OR (95% CI)
Gestational age at birth			
≥28 weeks	290	185 (63.8)	1.00 (Reference)
<28 weeks	78	36 (46.2)	0.49 (0.29–0.81)
Multiples			
No	253	150 (59.3)	1.00 (Reference)
Yes	115	71 (61.7)	1.11 (0.71–1.74)

**Table 5 ijerph-18-04140-t005:** Maternal demographics and MOM feeding at discharge.

Modifying Factor		MOM Feeding at Discharge	
	Total	N (%)	OR (95% CI)
Maternal age			
≥25 years	333	205 (61.6)	1.00 (Reference)
<25 years	35	16 (45.7)	0.53 (0.26–1.06)
Professional education			
No academic degree	263	159 (60.5)	1.00 (Reference)
Academic degree	51	36 (70.6)	1.57 (0.82–3.01)
Parity			
Primipara	222	135 (60.8)	1.00 (Reference)
Multipara	146	86 (58.9)	0.92 (0.60–1.41)

**Table 6 ijerph-18-04140-t006:** Infant morbidities and MOM feeding at discharge.

Modifying Factor		MOM Feeding at Discharge	
	Total	N (%)	OR (95% CI)
Intraventricular hemorrhage, grade 2 or higher			
No	345	211 (61.2)	1.00 (Reference)
Yes	22	9 (40.9)	0.44 (0.18–1.06)
Posthemorrhagic ventricular dilatation			
No	354	214 (60.5)	1.00 (Reference)
Yes	14	7 (50.0%)	0.65 (0.22–1.91)
Abdominal surgery			
No	349	214 (61.3)	1.00 (Reference)
Yes	19	7 (36.8%)	0.37 (0.14–0.96)

**Table 7 ijerph-18-04140-t007:** Maternal morbidities and MOM feeding at discharge.

Modifying Factor		MOM Feeding at Discharge	
	Total	N (%)	OR (95% CI)
Nicotine or recreational drug use			
No	332	214 (64.4)	1.00 (Reference)
Yes	36	7 (19.4)	0.13 (0.06–0.31)
Gestational diabetes			
No	331	205 (61.9)	1.00 (Reference)
Yes	37	16 (43.2)	0.47 (0.24–0.93)

**Table 8 ijerph-18-04140-t008:** Risk score and MOM feeding at discharge.

		MOM Feeding at Discharge	
	Total	N (%)	OR (95% CI)
MOM feeding risk score			
0	175	121 (69.1)	1.00 (Reference)
1	111	68 (61.3)	0.71 (0.43–1.16)
2	74	31 (41.9)	0.32 (0.18–0.57)
3	8	1 (12.5)	0.06 (0.01–0.53)

## Data Availability

The data presented in this study are available on request from the corresponding author. The data are not publicly available due to privacy reasons.

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
