# Peer review of "Mother’s Own Milk Feeding in Preterm Newborns Admitted to the Neonatal Intensive Care Unit or Special-Care Nursery: Obstacles, Interventions, Risk Calculation"

_ijerph, 2021, doi:10.3390/ijerph18084140_

Round 1
Reviewer 1 Report
The article named “Mother’s own milk feeding in preterm newborns admitted to the neonatal intensive care unit or special care nursery: obstacles, interventions, risk calculation” by Heller and colleagues explains about the different modifiable factors that can influence exclusive breastfeeding after the discharge of the newborn from the Neonatal Intensive Care Unit. It is a wonderful effort by the authors to put together factors that can be easily modified and can be easily followed by NICU’s throughout the world. Factors such as skin to skin kangaroo mother care soon after delivery, breastfeeding atleast once before discharge in the presence of a medical professional or lactation consultant when the neonate is in the hospital will help the parents establish relationship with the baby and acquire confidence in handling their babies. The authors have calculated the odds to show the techniques being beneficial to establish future long-term breastfeeding. The checklist at the time of discharge to improve feeding of Mother’s Own Milk (MOM) is comprehensive and easy to follow. Overall, this article can help NICU’s around the world increase their efficiency in discharging neonates on MOM. These are some minor revisions that are required.
- The authors have mentioned the gestational age of the infant in the “Infant Characterisctics and MOM eeding at discharge table. However, it would be interesting to know if the weight of the neonate influences breastfeeding similar to gestational age in their institution.
- Since this is an international journal, it would be better if WHO is mentioned before AAP in line 38-39.

Reviewer 2 Report
The study reviews published evidence on MOM feeding in preterm infants as well as presents own data on modifiable and unmodifiable factors of MOM feeding. There authors propose 9-point scale for non-MOM feeding of preterm infants.
I would like the authors to comment on:
- why did you analyse the data from 2013-2015, instead more recent data, as the promotion and education of breastfeeding is an ongoing process. Do you think that the rate of MOM-fed preterm infants in your center is the same?
- Could you discuss the rate MOM-fed intfants at discharge in your institution with other data
- Could you discuss strength and weakness of proposed risk-factors scale?
Reviewer 3 Report
This is a thorough summary of breast feeding practices in the NICU and the numerous obstacles breastfeeding encounters. It should be very useful for personnel working in NICUs.
One item that is missing is nutrient fortification. It not being mentioned at all could be interpreted to mean that the authors are not practicing fortification. If that were the case it would need a justification. If not, mentioning the start and end of fortification would be a minimum.
Tables: What is confusing is that the item "Total" is placed under the heading "MOM feeding at discharge" when in reality "Total" represents the number of subjects in whom this was assessed. A modification of the tables, or at least a note explaining this discrepancy, would be in order.
Specific comments:
- The term "corrected GA" is used sometimes and sometimes not. Was there a difference between corrected GA and plain GA?
- Line 40: change "production" to "utilization"
- Lines 85-86: The sentence would be more easily understood if it read: First SSC in the NICU was initiated later in the NMOM group at 7 (4; 12) days of life than in the MOM group, where it was initiated at 5 (3; 10) days of life (p < 0.02).
- Line 113: replace "instead" by "than in infants born after 28 weeks, where it was initiated at ..."
